# From Rapid Recommendation to Online Preference-Sensitive Decision Support: The Case of Severe Aortic Stenosis

**DOI:** 10.3390/medsci6040109

**Published:** 2018-11-29

**Authors:** Jack Dowie, Mette Kjer Kaltoft

**Affiliations:** 1Department of Public Health, Environments and Society, London School of Hygiene and Tropical Medicine, 15-17 Tavistock Place, London WC1H 9SH, UK; 2Department of Public Health, University of Southern Denmark, J.B. Winsløws Vej 9B, 5000 Odense C, Denmark; mkaltoft@health.sdu.dk

**Keywords:** aortic stenosis, SAVR, TAVI, GRADE, MAGIC, person-centred care, decision support, Multi-Criteria Decision Analysis, preferences

## Abstract

The launch of ‘Rapid Recommendations’ by the Grading of Recommendations Assessment, Development and Evaluation (GRADE) group, in collaboration with Making GRADE the Irresistible Choice (MAGIC) and the British Medical Journal (BMJ), is a very interesting recent development in e-healthcare. Designed to respond quickly to developments that have created new decision situations, their first project resulted from the arrival of minimally invasive Transcatheter Aortic Valve Implantation (TAVI) as an alternative to Surgical Aortic Valve Replacement (SAVR), for patients with symptomatic severe aortic stenosis. The interactive MAGIC decision aid that accompanies a Rapid Recommendation and is the main route to its clinical implementation, represents a major advance in e-health, for a cardiovascular decision in this case. However, it needs to go further in order to facilitate fully person-centred care, where the weighted preferences of the individual person are elicited at the point of decision, and transparently integrated with the best (most personalised) estimates of option performances, to produce personalised, preference-sensitive option evaluations. This can be achieved by inputting the collated GRADE evidence on the criteria relevant in the TAVI/SAVR choice into a Multi-Criteria Decision Analysis-based decision support tool, generating a personalised, preference-sensitive opinion. A demonstration version of this add-on to the MAGIC aid, divested of recommendations, is available online as proof of method.

## 1. Introduction

Until a few years ago, a 70-year-old with severe symptomatic aortic stenosis faced a typical life expectancy of 50% at two years, with escalating heart failure and reduced quality of life—unless they were at ‘low or intermediate’ surgical risk, in which case they had the option of open heart surgery for valve replacement (Surgical Aortic Valve Replacement or SAVR).

The arrival of Transcatheter Aortic Valve Implantation (TAVI), with delivery of the replacement valve through the femoral artery, created a new, minimally invasive option. The resulting TAVI or SAVR decision was the first addressed by the newly-launched collaboration between Grading of Recommendations Assessment, Development and Evaluation (GRADE), Making GRADE the Irresistible Choice (MAGIC), and the British Medical Journal (BMJ) (hereafter GRADE+) to produce and publish ‘trustworthy recommendations … in response to potentially practice changing evidence, so called ‘Rapid Recommendations’ [1] (p. 2).

In the systematic review and meta-analysis of randomized controlled trials undertaken as part of the Rapid Recommendation production process, it was confirmed that neither was a dominant option. Each was superior on some criteria and inferior on others.
Mortality was reduced with transfemoral TAVI compared with SAVR by about 3%, stroke by 2%, acute kidney injury by 5%, bleeding by 24%, new onset atrial fibrillation by 18%, and duration of index admission by three days. These benefits, however, come with associated harms. TAVI was associated with an increased risk of experiencing symptoms of heart failure by about 6% (2% of which were moderate or severe), permanent pacemaker insertion by about 15%, and aortic valve reintervention over the short term by about 1%.[1] (p. 8)

It was noted that TAVI was an increasingly popular alternative to SAVR, even when this was not in line with current practice guidelines. These recommended either TAVI or SAVR in patients at high surgical risk—defined as a Society of Thoracic Surgeons predicted risk of mortality (STS-PROM) score of 8% or less, but recommended SAVR over TAVI for lower surgical risk patients. Despite this, half of the TAVI centers in Europe were found to perform TAVI in intermediate-risk patients (STS-PROM 4–8%) and 10% of centers did so in low-risk patients (risk score <4%).

The questions of interest in this paper do not concern the quality of the evidence produced as the basis of the new Rapid Recommendations, nor the grading of that evidence. They relate to the form the recommendations take, and the way they were generated on the basis of the evidence, but especially to the framing and content of the interactive online MAGIC decision aid. This is presented as the main route to clinical implementation of the recommendations, whether in the clinic or in the formulation of clinical practice guidelines. MAGIC decision aids undoubtedly represent a major advance in e-health, in this case in cardiovascular medicine. However, the aids can be much enhanced by an add-on that transforms them into e-health decision support tools that meets key requirements of person-centred decision making. Such an add-on is introduced in a later section as proof of method.

## 2. Materials

The materials drawn on are the BMJ article introducing the Rapid Recommendations on TAVI vs SAVR [2], other supporting papers [1,3] and the online MAGIC app (https://app.magicapp.org/app#/guideline/1308). (It is stressed in all these sources that patients without symptoms, or with milder aortic disease, are not covered.)

### 2.1. The Rapid Recommendations

Separate recommendations were arrived at for four age groups. The short summary quotations following are from the BMJ online presentation [2].
Under 65: Strong recommendation for SAVR. ‘Since durability of TAVI valves is unknown, younger people may place a high value on avoiding a second aortic valve placement.’65 and under 75: Weak recommendation for SAVR. ‘People who wish to avoid open-heart surgery are likely to favour TAVI. People who place more value on avoiding a second aortic valve placement are likely to choose surgery.’75 and under 85: Weak recommendation for TAVI. Same as statement for previous group.85 and over: Strong recommendation for TAVI. ‘The uncertainty around long-term durability of TAVI valves is not likely to concern those over 85. These older patients are also likely to place a high value on avoiding open heart surgery.’

In summary, the ’age-stratified recommendations reflect that TAVI is probably preferable to those over 75 years old, whereas SAVR is likely preferable to those under 75 years’.

More elaborated statements appear in the decision aid in the MAGIC app (link above). For the 65–75 age group, for example:
We suggest SAVR rather than TAVI [Weak recommendation].Benefits outweigh harms for the majority, but not for everyone. The majority of patients would likely want this option.Preference and values: Patients are likely to place different value on benefits and harms associated with TAVI. Patients aged 75 or younger—with a life expectancy well beyond 10 years—are likely to place a particularly high value on avoiding need for a second aortic valve replacement and are likely to choose surgery. Patients who place a high value on avoiding initial open heart surgery and are willing to accept an increased risk for aortic valve reintervention are likely to choose TAVI. A systematic review of values and preferences provided limited evidence to inform our judgements.

GRADE classifies recommendations as either ‘strong’ or ‘weak’:
Strong recommendations mean that most informed patients would choose the recommended management and that clinicians can structure their interactions with patients accordinglyWeak recommendations mean that patients’ choices will vary according to their values and preferences, and clinicians must ensure that patients’ care is in keeping with their values and preferencesStrength of recommendation is determined by the balance between desirable and undesirable consequences of alternative management strategies, quality of evidence, variability in values and preferences, and resource use [4] (p. 1051).

How were the specific recommendations arrived at in the valve replacement case? The Evidence to Decision (EtD) framework [5] is the generic GRADE answer:
Panels should *consider* the evidence about the benefits and harms of the options and how certain that evidence is. They also need to *consider* how much the people affected directly by the decision value the benefits and harms, whether there is important uncertainty about this, and whether there is important variability in how much people value the benefits and harms. They must then *consider* all these criteria together to make a judgment about the balance between the desirable and undesirable effects of the option.[5] (p. 7, italics supplied)

In the present cardiovascular case, the account of the recommendation process provided is:
When moving from evidence to recommendations, the panel *integrated* information on benefits and harms of treatment alternatives, quality of evidence, and values and preferences of patients as well as acceptability, feasibility, and resources.[2] (p. 2, italics supplied)

The GRADE account is therefore limited to saying that the recommendations resulted from making judgments about the various elements of evidence and then integrating them by a process of consideration. However, the concern here is not so much with the process as with the product, and in particular, because of its centrality in person-centred care, with the treatment of the individual’s preferences. The existence of preference heterogeneity is acknowledged—as ‘variability in values’—but this leads not to the abandonment of recommendations, but to the ‘weakening’ of them:
Uncertainty about how much those affected (patients or their carers) value the outcomes of interest can be a reason to make a weak (conditional) rather than a strong recommendation. Variability in how patients value the main outcomes (to the extent that individuals with different values would make different decisions) is another reason for a weak recommendation. For example, some patients might place a lower value on avoiding a stroke compared with avoiding serious gastrointestinal bleeding or the burden of warfarin treatment than other patients.[5] (p. 4)

The suggestion (in an earlier quotation) that patients whose preferences are not in accord with those of ‘most informed people’ should, by implication, be ignored, and a strong recommendation implemented, is concerning. Does the MAGIC decision aid (link above), explicitly designed for use in clinical consultations, help ensure preference-sensitive decision making for *all* patients?

### 2.2. The MAGIC Decision Aid

The online MAGIC decision aid which accompanies the Rapid Recommendations is intended to be delivered by the clinician in a shared decision-making context:
This interactive tool for shared-decision making is designed to help you meet your patients’ needs by:
Exploring what outcomes they wish to discussCommunicating the benefits and harms of each alternative, as well as their (un)certaintyDiscussing practical issues associated with each alternativeThis decision aid does not replace clinical judgment. Adapt it to the context as needed and use your own communication style.

The essence of the aid can only adequately be conveyed by seeing its two main visual components. They are reproduced here in the belief that both this reproduction and the subsequent development of an adaptive add-on, are in accordance with the licencing:
This is an Open Access article distributed in accordance with the Creative Commons Attribution Non Commercial (CC BY-NC 4.0) license, which permits others to distribute, remix, adapt, build upon this work non-commercially, and license their derivative works on different terms, provided the original work is properly cited and the use is non-commercial.[2] (Footnote)

It contains the following statement along with an active link to the MAGIC app containing the reproduced figures:
For a fully interactive version of this graphical summary of recommendations, please visit: https://doi.org/10.1136/bmj.i5085

The Disclaimer to the MAGIC infographic is also important to note, both substantively and because it also applies to the future adaptation of the aid, undertaken as proof of method:
Disclaimer: This infographic is not a validated clinical decision aid. This information is provided without any representations, conditions or warranties that it is accurate or up to date. BMJ and its licensors assume no responsibility for any aspect of treatment administered with the aid of this information. Any reliance placed on this information is strictly at the user’s own risk.

Figure 1 is the summary MAGIC infographic for the 65–75 age group. Figure 2 is the matching picture of the full set of individual evidence boxes, envisaged as being selected, loaded, and discussed one-by-one during a shared decision-making sequence.

## 3. Method

The questions to be asked at this point are why the recommendation is needed in the first place if personalised decision support is being offered, and why it is displayed prominently at the top of the infographic as part of the decision aid (Figure 1). Rhetorically, can a genuine decision aiding process take place within the frame of a recommendation for the decision, even if it is ‘weak’?

Apart from this problematic framing, the impressive infographic presented as part of a MAGIC decision aid, may paradoxically increase the difficulty of processing the information *into a decision*, simply because it is set out so clearly and attractively. GRADE+, albeit in good company, holds a highly optimistic view of the cognitive ability of humans (including health professionals) to synthesise even the modest quantities of disparate information in this infographic into an accountable decision, i.e., a decision for which even a modestly explanatory account can be provided. This optimism is associated with the belief that communicating information as *information*, in a well-organised and attractively laid-out form, is both empowering and constitutes effective *decision* support.

In contrast, such a presentation of information can easily be seen as overpowering, not empowering, possibly even as constituting ‘symbolic violence’ [6]. Only communicating information within an explicit decision framework constitutes *decision support*, as opposed to *information support for decision making*. Visually, a ‘decisiographic’, not an infographic, is needed for this task. It is argued elsewhere that online decision support tools based on Multi-Criteria Decision Analysis (MCDA) are good candidates for person-centred decision making and that the Annalisa implementation of MCDA provides a simple ‘decisiographic’, displaying evidence, preferences, and the scores which result from their integration on a ‘one-screen-shows-all’ [7]. A screen capture appears in a later section (Figure 3) introducing the decision support add-on to the MAGIC aid.

Person-centred care requires the preferences of the person-as-patient to be elicited at the point of decision, and transparently integrated with the best (most personalised) available estimates of option performances, to produce preference-sensitive option evaluations. The lack of population-level evidence on the ‘balance of benefits and harms’, regretted by the present recommendation developers (‘A systematic review of values and preferences provided limited evidence to inform our judgements’.) is not a problem, and the further research called for is not needed. Group average preferences, such as those derived in Discrete Choice Experiments or Conjoint Analyses, are not relevant in person-centred care [8,9] and their continued claims of clinical relevance are examples of the methodological research tail wagging the clinical practice dog. It is the harm–benefit balance of the individual making the TAVI–SAVR decision that needs to be obtained and applied, at the point of decision.

Such individualised support is also arguably essential for the giving and obtaining of informed consent under the recent Montgomery ruling in the UK, which substituted the ‘reasonable patient’ for the ‘reasonable practitioner’ standard [10]. The reasonable patient will certainly want to know how well their options perform on the considerations that matter to them, considerations which they themselves weight in importance.

The required decision support tool must be practical and useful. It must meet the SMART criteria—Specific, Measurable, Achievable, Relevant, and Timely—to the extent that these are reasonable in the specific decision context. More simply, the decision support must be ‘fast and frugal’. It must be pitched at the most appropriate point (trade-off) on the ‘rigour-relevance’ continuum and be cost-effective as a contributor to the quality of the decision. It will therefore be a long way from a highly analytical and quantitative evaluation of the type undertaken by the UK National Institute for Health and Care Excellence, but equally distant from an expert intuition only-based verbal deliberation or evaluation in the otheropposite direction.

Multi-Criteria Decision Analysis (MCDA) as implemented in Annalisa [7] can provide the basis for such ‘fast and frugal’ decision support tools. They provide an opinion on the worth of each Option by combining its Ratings (how well it performs on relevant Criteria) with its Weightings (how important is a criterion relative to the other criteria), making the opinion a preference-sensitive one. Introductory materials and numerous examples of the implementation of the technique are available at http://cafeannalisa.org.uk, including short video ‘Powtoons’. These draw on the increasingly popular comparison websites for goods and services, such as *Which* (UK), *Tænk* (Denmark), *Consumer Reports* (US), and *Choice* (Australia), all of which employ a basic and accessible form of MCDA. Examples taken from them are useful ’warm-ups’ to health applications.

Alan Williams lamented attempts to escape from the daunting demands of decision making by engaging only in deliberative consideration based on verbal judgments, rather than requisite quantification and calculation. As he argued, whether in guideline formation panels or clinical consultations, ‘consideration’ unsupported by coherent quantitative analysis lacks transparent accountability and hence provides ample cover for bias and discrimination. Adopting this position, he was prepared for ‘the hostility it is likely to engender from those who mistakenly equate precision with lack of humanity’ [11] (p. 120). This is also possible in relation to the current proposal.

## 4. Result: An Add-On Decision Support Tool for the TAVI versus SAVR Choice

The basic inputs into the decision support tool are seven featured criteria in the MAGIC decision aid, along with the summary evidence on how TAVI and SAVR perform on them, as their Ratings. Three considerations, where the individual is the expert, are added and they are asked to supply their personal performance ratings, based on the relevant information taken from the aid, including its ‘Practical Considerations’ section. These 10 become the criteria in the interactive decision support tool, for which the person provides a set of criterion importance Weightings. In line with the principles of value-based compensatory MCDA, the Weightings and Ratings are then integrated in expected value calculations to generate a preference-sensitive *opinion*—pair of Option Scores—for the person to discuss. They can revise their Weightings in the light of the displayed Ratings. (The Ratings for the seven MAGIC criteria are ‘locked’ so as to prevent editing and possible misrepresentation.)

A sample of the Annalisa output screen appears in Figure 3. This example, for an under 65 year old, is one where a reasonable set of criterion importance Weightings and respondent-sourced performance Ratings produces an opinion favouring TAVI—in contrast to the strong SAVR recommendation for this age group.

GRADE uses four verbal levels to classify the quality (‘certainty’) of the evidence. In our conception of a decision support tool, adjusting for this is not a task to be left outside the tool for ‘consideration’. MCDA requires quantitative inputs, so the GRADE certainty levels are mapped on to the 0–1.0 scale: Very low = 0.1; Low = 0.4; Moderate = 0.7; High = 1.0. (These particular mappings are open for debate, but the principle of numerical mapping is not.) The pair of certainty-adjusted Scores for TAVI and SAVR is presented in the aid alongside the unadjusted ones, so that the effect of certainty adjustment is clearly visible.

To engage with the tool, on a demonstration-only basis, go to https://ale.rsyd.dk and enter 1491 as survey ID. Feedback is invited.

While offering substantive support for this particular decision, any engagement with this sort of tool is likely to have spillover empowerment benefits in the form of enhanced health decision literacy—and indeed generic decision literacy—through the introduction to MCDA it provides.

## 5. Discussion

The first thing to note is that there has been interference with person-centred care in the pathway that has led to the TAVI or SAVR decision node being reached. The Recommendation does not question the restriction of the SAVR option to persons at ‘low or intermediate’ surgical risk. Recommending an 8% threshold for surgical mortality risk may be in providers’ interest to preserve their place in some league table, or professional morale and job satisfaction, but it infringes a key principle of person-centred care [12]. A patient should be free to decide between TAVI, SAVR, and a life expectancy of three years with much reduced quality of life, whatever their surgical risk. What they need is preference-sensitive support in making such a tough decision. In fact, all population-based threshold-based risk classifications threaten person-centred care and good clinical practice, by interfering with obtaining the legally, and ethically, required informed and preference-based consent [13,14].

Where does this leave GRADE and GRADE+? Person-centred care requires the individual person’s preferences to be elicited at or near the point of decision, so that their decisional autonomy is respected, and so that their legal informed consent can be obtained to any provider action following from a decision.

Decision support is needed in such a complex decision, but the support must not be biased by being framed within a recommendation provided, or not provided, on the basis of which of two categories the evidence fits according to a guideline panel. The TAVI or SAVR decision is a near-perfect example of why this should be avoided, and why the under 65 s and over 85 s should be offered exactly the same support as those between those ages. The opinion emerging from the add-on decision support tool correctly reflects the age-specific evidence for death and stroke, but is not ‘ageist’ in any other way.

The possibility of decision aids being useful is acknowledged in GRADE, but conditionally:
As clinicians become more aware of variability in patients’ values and preferences, they are turning to structured decision aids to facilitate the decision making process. A strong recommendation indicates that use of a decision aid is unnecessary—almost all informed patients will make the same choice. A weak recommendation indicates that a decision aid could be useful.[4] (p. 1049)

On the contrary, in person-centred healthcare, a preference-sensitive, decision analytic support tool should always be offered. If practice guidelines are to survive in this context, the relationship between guidelines and decision aids needs to be the reverse of that suggested by van der Weijden, et al., where the guidelines are developed first and the decision aids derived from them later [15]. While they note the disconnect between the groups responsible for guideline and aids, and suggest greater collaboration is the answer, they overlook the fundamental distinction between establishing a good evidence base in healthcare *research* and making preference-sensitive decisions in healthcare *practice*.

The important function of groups such as GRADE and MAGIC is to develop and present the evidence base in the light of the structure of the decision aid, eschewing the issue of recommendations which necessarily involve preference judgments. It is particularly worrying when authors appear to feel entitled to decide whether decisions are ‘preference-sensitive’ or not. All decisions are preference-sensitive, even if all agree on the preferences involved in a decision. If a proxy is needed, the proxy should be using the preferences they believe the patient holds, not the ones they think the patient *should* hold. The evidence should be presented in a personalised and preference-sensitive decision support tool, uncontaminated by any prior assumptions about sensitivity.

While the argument made here may seem radical, in most respects it merely echoes the views of Margaret McCartney and colleagues [16]. Here are some selected excerpts, but their whole paper is essential reading:
… there is the danger of guideline recommendations being applied to people who do not place the same values on those recommendations as their clinician, or indeed those intended by the guideline creators… Surveys have shown that most patients wish either to share decision making with their clinicians or to take the decisions themselves. Guidelines should enable, not subvert, this process… guideline producers need to resist the temptation to tell clinicians and patients what to do. Making recommendations for the population, often based on expert opinion, reinforces the power imbalance between professional expertise and the patient’s values and preferences… The impact of the side effect at the level of the population may be high if the treatment is used widely, and may be considered important enough to withdraw the treatment from general use. This removes a potentially valuable treatment option for someone who has perhaps found the treatment highly effective and has a very low absolute risk of being harmed…It is difficult to personalise recommendations from guidelines, even for those skilled in evidence based medicine… Usable decision aids should [therefore] now be seen as one of the most important end products for evidence based medicine.[16] (pp. 1–2)

In keeping with GRADE, MAGIC, and the BMJ, the aim here is to provide support for more transparent and accountable decisions, made within typical time and practice constraints and cognitive limitations of all parties. Our own decision support tools, intended for apomediative (at home) use, as well as in subsequent clinical encounters [17,18], do not seek to meet the standard normative checklist for decision aid development, produced without any notion of timeliness or cost-effectiveness [19]. Paradoxically, setting these longer, more expensive and demanding parameters for decision aids can have the effect of encouraging poorer, analytically unsupported, decision making. Any specific decision support tool must be evaluated empirically and comparatively and only against the actual decision-making process that would be followed in its absence. The normatively best is the enemy whenever it discourages the development and use of the empirically better. In this respect, the current add-on tool is not a good example, since it builds on an aid that does meet the demanding standard requirements.

The limitations of the current proposal are all those associated with attempts to introduce decision aids into clinical practice [20,21], perhaps increased by its grounding in the MCDA approach to decision support. This approach is currently seen as difficult to reconcile with conventional models of clinical reasoning. However, since it is proving difficult to introduce patient preferences explicitly and transparently into these models, the MCDA approach is making slow progress towards acceptance. Whether this is a limitation is therefore a function of one’s assessment of the state of current practice.

Implementing the type of decision support proposed here as an add-on to the MAGIC decision aid would have significant clinical implications; the more so the greater the role guideline recommendations play in actual clinical care. Most guideline recommendations are based on evidence regarding the main benefit criterion in a decision and on the perceived average preferences of patients. Person-centred care in a heterogeneous population requires that such recommendations should be interpreted in the light of a wider multi-criteria analysis of all the benefits and harms associated with the individual patient’s options, as assessed using their personal preferences. While this may be seen as required in ethical and legal practice, there are serious doubts as to the extent to which it is delivered within the pressures of practice. Concerns about the extra clinical time and resources needed to implement this approach to decision support are best met by considering its introduction apomediatively, through e-health and m-health technologies.

## 6. Conclusions

Whether or not the widely-disseminated MAGIC aid addresses the full complexity of the valve replacement decision, as viewed by patients and revealed in qualitative studies [22], is left to others to decide. In an original decision support tool, as opposed to this add-on, other criteria might be added, along with the mandatory ‘do nothing’ option. However, this add-on to provide preference-sensitive decision support is seen as a response to the GRADE+ invitation encouraging ‘adaptation of recommendations to allow contextualisation’ in pursuit of improved healthcare. Feedback is welcomed.

## Figures and Tables

**Figure 1 medsci-06-00109-f001:**
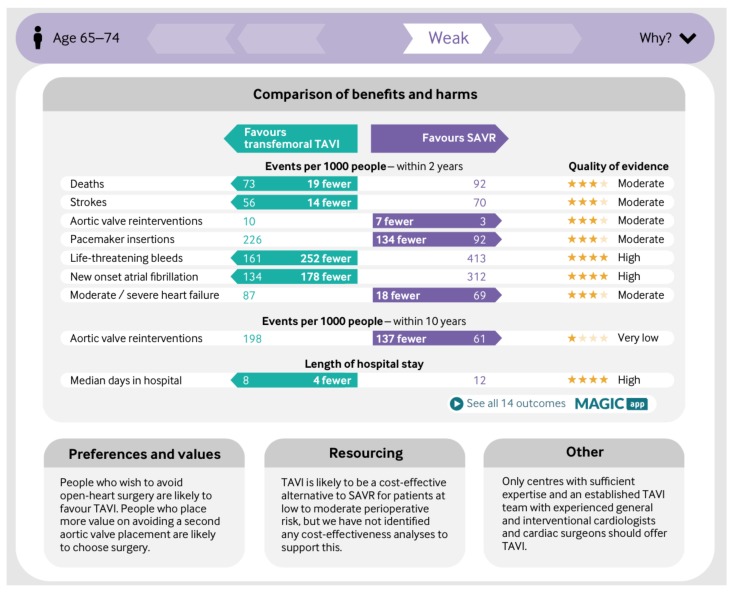
Infographic in Making GRADE the Irresistible Choice (MAGIC) aid for 65–75 year-olds.

**Figure 2 medsci-06-00109-f002:**
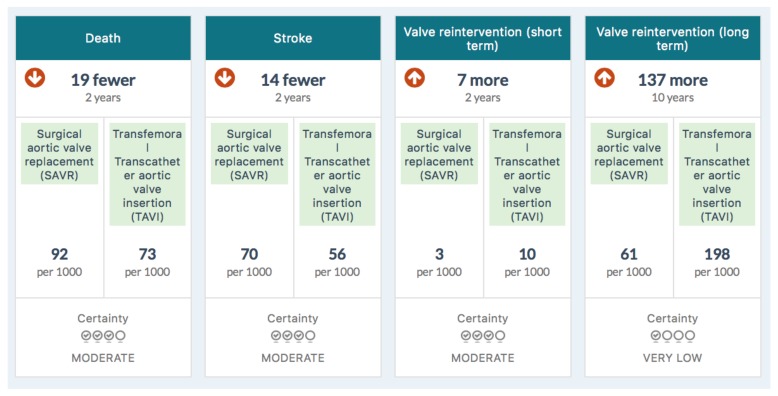
Information boxes in the MAGIC aid for 65–75-year-olds.

**Figure 3 medsci-06-00109-f003:**
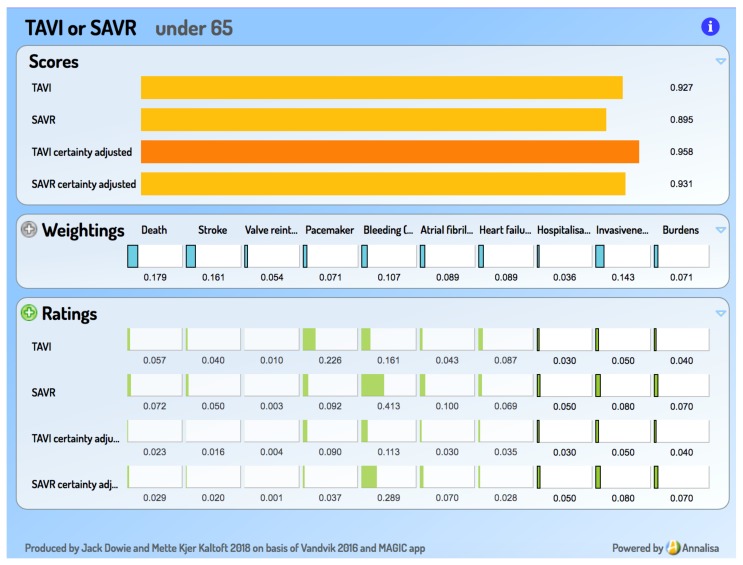
Screen capture from the MAGIC add-on decision support tool.

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
