# Peer review of "From Rapid Recommendation to Online Preference-Sensitive Decision Support: The Case of Severe Aortic Stenosis"

_medsci, 2018, doi:10.3390/medsci6040109_

Round 1

Reviewer 1 Report

There are two fundamental problems in our current clinical practice: we are market driven and we rely too much on guidelines, which should be considered only as such because they are not "gospel". As long as a decision aid/tool is considered in the context of the clinical picture and is not meant as a substitute of the clinician's experience and expertise, then it may well be a useful and welcome approach. The article addresses a current hot topic: TAVI is not the "magic bullet" and it is not free of complications. I maintain my reservations for its indication in younger patients or so called "low- or medium-risk patients". Surgery still plays a role in high-risk patients. Finally, the publication of surgeon-specific results has affected current practice generating self-preservation and risk-averse behaviour with significant impact on decision-making. Paradoxically, the higher-risk patients are those who most likely may benefit from surgery provided that the whole context is taken into account. This is a timely article, which I hope may well generate further debate in the field. I would strongly suggest the authors to persevere with their efforts. 

Author Response

We are very pleased that the reviewer accepts the importance of the argument made in the paper, which is indeed to ensure that decision support is personalised, transparent and complementary to clinical expertise. We do intend to pursue - and persevere with - this work.

Reviewer 2 Report

A decision support tool must be practical and useful and this add-on to provide preference-sensitive decision support.

Taking into account the principle of person-centred care, a patient should be free to decide between TAVI, SAVR and a life expectancy of three years with much reduced quality of life, whatever their surgical risk, with the treatment of individual’s preferences.

The structure of the document is unclear. It would improve the clarity with a classic structure of introduction, Materials and methods, Results, Discussion and conclusions.The reader would know where it is in each moment of the reading with a classic structure.

Author Response

We are very pleased that the reviewer accepts the importance of the argument of the paper. We also agree that using the classic structure will help the reader follow the steps in that argument and have introduced it in this revision.